# Molecular-Level Insight of CP52/NBR Damping Composites through a Combination of Molecular Dynamics Simulation and Experimental Method

**DOI:** 10.3390/polym15030604

**Published:** 2023-01-24

**Authors:** Qingxin Wang, Yu Li, Shuaijie Li, Zhaoyi Sun, Guorong Wang

**Affiliations:** 1Department of Basics, Naval University of Engineering, Wuhan 430033, China; 2Troops 91411 of Chinese People’s Liberation Army, Dalian 116000, China; 3College of Naval Architecture and Ocean, Naval University of Engineering, Wuhan 430033, China

**Keywords:** nitrile butadiene rubber, chlorinated paraffin 52, damping property, olecular dynamics simulation, analogous hydrogen bond

## Abstract

To enhance the damping properties of nitrile butadiene rubber (NBR), the elastomer used was blended with chlorinated paraffin 52 (CP52) to prepare NBR/CP52 composites. The results showed that CP52 could significantly enhance the damping properties of NBR and shift the glass transition temperature (*T*_g_) to lower temperatures. Molecular dynamics models of the CP52/NBR system were established, and the damping properties of the CP52-reinforced NBR were investigated using molecular dynamics (MD) simulations. Through the combination of MD simulations and the experimental results, the essential mechanism of the enhanced damping properties of the NBR was methodically expatiated and was ascribed to the Cl-CP-H····NC-NBR (type I) and CP-Cl····H-NBR-CN (type II) analogous hydrogen bonds formed between NBR and CP52. The higher the CP52 content, the higher the analogous hydrogen bond concentration, and the better the damping properties of the CP52/NBR composites. The experimental results were very consistent with the MD simulation results, meaning that the combination method can provide a new means to optimize the design of damping materials and broaden the application range of small polar molecules in the damping modification of polar rubber materials.

## 1. Introduction

Currently, vibration and noise problems are becoming increasingly prominent, and the application of damping materials is one of the most direct and effective methods for reducing vibration and noise [1]. Nitrile butadiene rubber (NBR) has excellent mechanical strength, damping properties, promising oil and heat resistance, and is widely used in damping materials [2,3]. However, due to the influence of the glass transition temperature (*T*_g_), the single-component NBR’s high damping temperature range is lower than room temperature, and the effective damping temperature range is generally narrow and cannot necessarily meet the needs of different engineering fields and application environments [4,5].

In general, the dynamic mechanical behavior of polymers can be adjusted by conventional means such as the additions of plasticizers and fillers, mechanical blending of various polymers, and copolymerization. However, the existing damping modification methods and research methods have certain limitations. For example, both blending and copolymerization can broaden the effective damping temperature range of rubber-based damping materials to a certain extent, but the blending materials have problems such as easy phase separation of different components and unstable mechanical properties. Copolymerization also has problems such as complicated preparation process, high batch production cost, and serious environmental pollution.

Organic small molecule hybrid damping material is an emerging damping material, which has attracted the attention of many researchers due to its excellent damping characteristics and special damping mechanism. At present has formed a relatively complete system, which is representative of hindered phenol, hindered amine and other organic small molecules containing polar functional groups. When a large number of sterically hindered or sterically hindered amine organic small molecules (content even greater than the polymer matrix content) are added to the polymer matrix, the prepared damping material exhibits high damping characteristics, and the damping peak position is adjustable, its performance is much better than the traditional damping material. The disadvantage of organic small molecule hybrid damping material is that the properties is unstable, and frosting will occur after long-term use, which will reduce the damping properties.

In the study, our team found that chlorinated paraffin (CP) can significantly enhance the damping properties of NBR, and no frosting occurs, the damping properties is stable. Chlorinated paraffin (CP) is a chlorinated derivative of saturated C_10~30_ straight-chain n-paraffin hydrocarbons [6]. According to different chlorine contents, chlorinated paraffin can be divided into CP52 and CP70 brands. Generally, paraffin with high n-paraffin content is used as a raw material and is prepared by the thermal chlorination method. It is usually added to rubber materials as a flame retardant or plasticizer, and the added amount is generally no more than 10 phr (parts per hundred of rubber) [7,8]. There are a few reports on using CP to enhance the damping property of rubber materials [9,10].

In this study, NBR was modified with the small polar molecule CP52 by using the organic small-molecule hybridization process to prepare CP52/NBR composites, and the effect of CP52 on the damping properties of NBR was studied. Using molecular dynamics (MD) simulations, the CP52/NBR system model was constructed, and the intermolecular interaction parameters were calculated. The relationship between the damping performance and the intermolecular interaction parameters was qualitatively analyzed. This study aimed to explore the mechanism of CP52 in improving the damping properties of NBR and provide theoretical support and technical guidance for subsequent related research, broadening the application range of small polar molecules in the damping modification of polar rubber materials.

## 2. Materials and Methods

### 2.1. Materials

NBR (4155) with an acrylonitrile mass fraction of 41% was provided by Nnatex Industry Co., Ltd. (Zhenjiang, China). CP52 was provided by the Damao Chemical Reagent Factory. (Tianjin, China). magnesium oxide (MgO), stearic acid (SA), DM (2,2′-dithiodibenzothiazole) and TMTD (tetramethyl thiuram disulfide) was provided by yuanhao Co., Ltd. (Hebi, China). All of the other raw materials are commercially available industrial products.

### 2.2. Preparation of CP52/NBR Composites

First, the NBR was kneaded with a mixing roller for 5 min, and CP52 was added to the kneaded NBR, which was then kneaded at room temperature for 5 min. Then, the rubber additives were added to the above samples, including 5 phr of MgO, 1 phr of SA, 1.5 phr of accelerator DM, 0.5 phr of accelerator TMTD, and 0.5 phr of sulfur. Evenly mixing for 10 min. Finally, the samples were hot-pressed and vulcanized for 15 min at 12 MPa and 150 °C and then cooled naturally to room temperature to obtain the specimens.

### 2.3. Characterization

Dynamic mechanical analysis (DMA, DMA1, METTLER TOLEDO, Zurich, Switzerland) was used to test the dynamic mechanical properties in the tensile mode. The test frequency was 10 Hz, the test temperature range was −60 °C~80 °C, and the heating rate was 3 °C/min.

Fourier-transform infrared spectroscopy (FTIR, Nicolet IS20, Thermo Fisher Scientific, Madison, WI, USA) was used to test the samples pre-pressed into films using the ATR module and total reflection mode. The scanning wave number ranged from 400~4000 cm^−1^.

### 2.4. Model and Simulation Details

The Materials Studio (MS) 2019 software was used to reveal the effect of CP52 on the NBR composites from the microscopic aspect, providing theoretical guidance for the experimental study of damping materials.

The MD simulation was carried out by using the Forcite and Amorphous Cell modules. During the simulation, an Andersen thermostat was used for temperature control, a Berendsen barostat controlled the pressure, and the condensed phase optimized molecular potentials for atomistic simulation studies (COMPASS) force field was adopted.

Figure 1 shows the construction procedure for the CP52/NBR composite amorphous cells. The acrylonitrile repeat units (Figure 1a) and butadiene repeat units (Figure 1b) were first randomly copolymerized as an NBR polymer chain (Figure 1c) with an acrylonitrile mass fraction of 41%. Then, the NBR polymer chains and CP52 small molecules (Figure 1d) were built in a periodic boundary cell (Figure 1e); according to the mass ratios, each cell consisted of 4 NBR chains with 50 repeat units and different CP52 molecules and are listed in Table 1. Subsequently, the cell’s energy was minimized by using the Smart Minimizer method to relax the state of the minimal potential energy. After the completion of energy minimization, in order to eliminate the internal stress, the cell was annealed from the low temperature of 200 K to the upper temperature of 400 K with 100 cycles to prevent the system from being trapped in a local high-energy minimum. Finally, the dynamics simulation was performed using an NVT ensemble (constant atomic number, volume, and temperature) at 298 K for 500 ps and 1000 ps of NPT (constant number of particles, pressure, and temperature). The simulation was carried out at 0.1 MPa to relax the polymer structure further. Then, the relevant physical parameters required for the system to reach perfect equilibrium were calculated.

The reliability of the model was tested in relation to energy and density. When the energy fluctuation in the system was less than 10% of the total energy and the density fluctuation was within 3%, the system was considered stable. The simulated density of the system and the density of the real sample are listed in Table 2. Compared with the experimental data, the simulated density deviates from the density measured in the real sample by no more than 3%. It shows that the model is close to the actual situation and can reflect the real performance of the material. The relevant data calculated by this model have reference value and can be used to analyze the damping mechanisms of CP52/NBR materials.

## 3. Results and Discussion

### 3.1. Effect of CP52 on the Damping Property of NBR

The effect of CP52 on the damping properties of NBR is shown in Table 3 and Figure 2. With the increase in CP52 content, the loss of the tangent (tan*δ*) and the peak area (*TA*, tan*δ* > 0.3) of the CP52/NBR composites increased, the effective damping temperature range (Δ*T*, tan*δ* > 0.3) broadened, and *T*_g_ shifted to a lower temperature. When the added amount of CP52 is 100 phr, the Δ*T* of the composite is 46.8 °C, and the tan*δ*_max_ is as high as 1.78, which is 62% higher than NBR without CP52. It shows that CP52 can significantly enhance the damping properties of NBR.

### 3.2. FTIR Analysis of CP52/NBR Material

The position and intensity of the absorption peaks on the FTIR spectra can reflect the state of the functional groups inside the material and the intermolecular forces. Figure 3a is the FTIR spectra of NBR and CP52 and the CP52/NBR materials, wherein (1) and (2) are the characteristic absorption peaks of -CN and -Cl. It can be found that, with increased CP52 content, the peaks are red-shifted [11], and the stretching vibration peak of -CN gradually changes from 2236 cm^−1^ to 2234 cm^−1^. The characteristic absorption peak of -Cl also moves to a lower frequency and continues to increase. The reason is that the strong induction effect leads to the averaging electron cloud distribution of the polar groups, causing the chemical bond between the polar groups and the main molecular chain to elongate, decreasing the wave number of the characteristic peak [12,13].

Therefore, it can be speculated that, based on the crosslinking network established by vulcanization, CP52 and NBR molecular chains can also form a network structure through intermolecular interactions. As shown in Figure 4, the thick black line represents the NBR molecular chain segment, the thin blue line represents the CP52 molecule, and the Cl-CP-H····NC-NBR and CP-Cl····H-NBR-CN intermolecular interactions are represented by imaginary lines as type I and type II, respectively. The interaction between these two molecules ensures the formation of a spatial network structure but is not as stable as a covalent bond. When subjected to alternating forces, such as vibration, the intermolecular interactions are continuously dissociated and generated; the mechanical energy is rapidly dissipated; thus, the damping performance of the CP52/NBR material improves.

### 3.3. Radial Distribution Function Analysis

The radial distribution function (RDF) refers to the radial distribution of a particle centered with respect to other particles, which can reflect the degree of order within the polymer molecules and intermolecular interactions [14]. As Figure 5 shows, (a) is the RDF spectra between the nitrile group on NBR and the para-hydrogen of the chlorine atom on CP52, while (b) is the RDF spectra between the chlorine atom on CP52 and the para-hydrogen of the nitrile group on NBR; corresponding to the type I and type II intermolecular forces proposed in the previous section, respectively. It can be seen that the RDF spectra of both forces have a peak near 3.0 Å, which proves that these two interactions exist in the CP52/NBR system, which is consistent with the FTIR test results.

According to reports [15,16], different intermolecular interaction forces form different probability densities. The type of intermolecular interaction can be determined by analyzing the morphology and peak position of the RDF spectrum. Generally, hydrogen bonds are distributed in the range of 2.6 Å to 3.1 Å [17]; strong van der Waals forces are between 3.1 Å and 5.0 Å [18], and weak van der Waals forces are more than 5.0 Å. The peaks of Cl-CP-H····NC-NBR and CP-Cl····H-NBR-CN are wide, near 2.9 Å and 2.8 Å, and the peak’s shape is influenced by some hydrogen bonds and strong van der Waals forces. The strength of the interaction is close to that of hydrogen bonds, and the bonding mechanism is similar to that of hydrogen bonds. Therefore, their intermolecular interaction is defined as an analogous hydrogen bond, where Cl-CP-H····NC-NBR is a type I analogous hydrogen bond, and CP-Cl····H-NBR-CN is a type II analogous hydrogen bond. Their structures are shown in Figure 6.

The number of the two types of analogous hydrogen bonds in the periodic cells and the molar concentrations of the analogous hydrogen bonds in the last 50 conformations of each system were calculated. As Figure 7 shows, it can be seen that, with the increased CP52 content, the mass fraction of NBR decreased, the concentration of the type I analogous hydrogen bonds in the system increased, and the type II analogous hydrogen bonds tended to decrease. However, the total concentration of the analogous hydrogen bonds (*C_I+II_*) increased.

It was found that the variation trend of *C_I+II_* is consistent with the measured tan*δ*_max_ and *TA*. To further determine the relationship between the molar concentration of the analogous hydrogen bonds and the damping property of the material, the above data were analyzed using the linear regression method. The regression curve is shown in Figure 8. The concentration of the analogous hydrogen bonds in the CP52/NBR materials is closely related to the damping properties. The correlation coefficient between *C_I+II_* and tan*δ*_max_ is 0.891, and the correlation coefficient with *TA* is 0.951.

It is inferred that the analogous hydrogen bond is the main factor affecting the damping performance of the CP52/NBR material. The plasticity of CP52 makes the NBR molecular chain segment easier to move under the action of external forces. The chain segment is continuously dissociated and generated in the movement process, and the external energy is continuously consumed, resulting in a good damping effect. The higher the CP52 content, the higher the total concentration of analogous hydrogen bonds, so the better the damping performance.

### 3.4. Binding Energy Analysis

When the system contains two or more components, the components are combined through intermolecular interactions, and the energy consumed to separate the components is the binding energy (*E_binding_*). *E_binding_* is defined as the negative value of intermolecular interaction energy (*E_binding_* = −*E_inter_* = −(*E_total_* − *E_CP52_* − *E_NBR_*)) [19]. *E_binding_* can characterize the strength of the intermolecular interaction. The larger *E_binding_* is, the stronger the interaction between the components in the system and the better the compatibility. The energy-related parameters of the CP52/NBR system are listed in Table 4. All of the *E_binding_* values are positive and enlarged gradually with the increased CP52 content, indicating good compatibility between CP52 and NBR and a more potent intermolecular force; this is because the analogous hydrogen bonds are formed between CP52 and NBR. The higher the CP52 content, the higher the analogous hydrogen bond content in the system, *E_binding_* is more prominent, and more external energy is consumed in the dissociation process. Therefore, it can be determined that the material has better damping performance.

### 3.5. Fraction Free Volume and Cohesive Energy Density Analysis

The fraction free volume (*FFV*) represents the percentage of free volume in the system’s total volume, which can reflect the tightness of the molecular accumulation in the system. Cohesive energy refers to the total energy that 1 mol molecules gather or the energy consumed to overcome intermolecular forces and transfer a molar of aggregated matter outside the range of the intermolecular forces [20]. Cohesive energy per unit volume is called cohesive energy density (CED) [21]. Generally, the changing trend of *FFV* and CED is the opposite [22]. The higher CED is, the stronger the intermolecular force, the closer the molecular stacking in the system, the smaller the *FFV*, and the lower the *T*_g_ of the material [23,24]. The calculation results of *FFV* and CED of the CP52/NBR system are shown in Figure 9. It can be seen that the changes in *FFV* and CED are not noticeable. With the increased CP52 content, *FFV* is always stable at around 38%, while CED fluctuates in the range of 330~380 J/cm^−3^. The changing trend regarding *FFV* and CED is roughly the opposite. Therefore, *FFV* is not the main factor affecting the damping property and the *T*_g_ of the CP52/NBR materials. Increasing the CP52 addition does not significantly change *FFV*. It is mainly due to the CP52 having a small molecular weight and filling in the macromolecular chains of NBR, making its chain segments move more easily, resulting in the glass transition region moving to a lower temperature.

## 4. Conclusions

In this article, CP52/NBR composites were prepared experimentally, and the effect of CP52 on the damping properties of NBR was studied. The molecular model was constructed to calculate the intermolecular interaction parameters of the CP52/NBR system. Then, the damping mechanism of the CP52/NBR composites was analyzed using the experimental results. The conclusions are as follows:(1)With increased CP52 content, the damping properties of the CP52/NBR composites were greatly improved, and the characteristic absorption peaks of -CN and -Cl on the infrared spectrum were red-shifted, indicating that intermolecular force could be formed between CP52 and NBR.(2)The RDF simulation results and FTIR test results show that there are two kinds of hydrogen bonds in the CP52/NBR damping materials, Cl-CP-H····NC-NBR and CP-Cl····H-NBR-CN. The linear regression analysis and *E_binding_* results showed that hydrogen bonding was the main factor affecting the damping properties of the CP52/NBR materials.(3)The *FFV* and CED of the CP52/NBR system did not change significantly with increased CP52 content, and the changing trend was roughly the opposite. It is indicated that *FFV* is not the main factor affecting the damping performance and *T*_g_ of the CP52/NBR system. The effect of CP52 on the glass transition region is mainly attributed to its plasticity.

On the basis of this research, MD simulation can be used to establish other polar small molecule and polar rubber system models, and experimental verification can be carried out to develop a series of damping materials with hydrogen bond network structure. Moreover, according to the principle of hydrogen bond-like, from the perspective of molecular structure and functional group, some polar small molecule substances were designed and synthesized for simulation and experiment, which further verified the ‘hydrogen bond-like theory’ and obtained excellent damping materials.

## Figures and Tables

**Figure 1 polymers-15-00604-f001:**
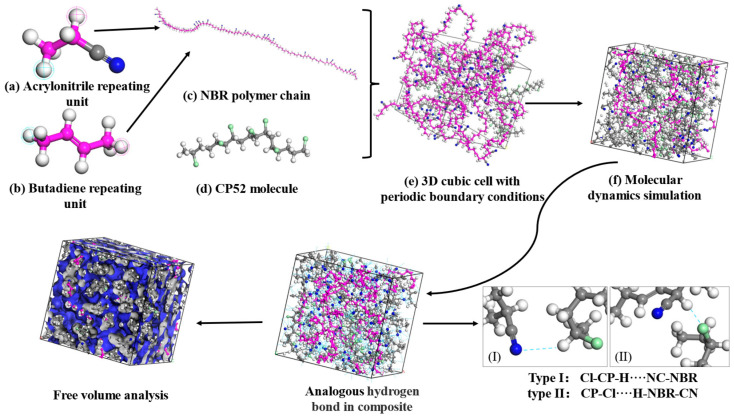
Models for molecular dynamics simulation of CP52/NBR.

**Figure 2 polymers-15-00604-f002:**
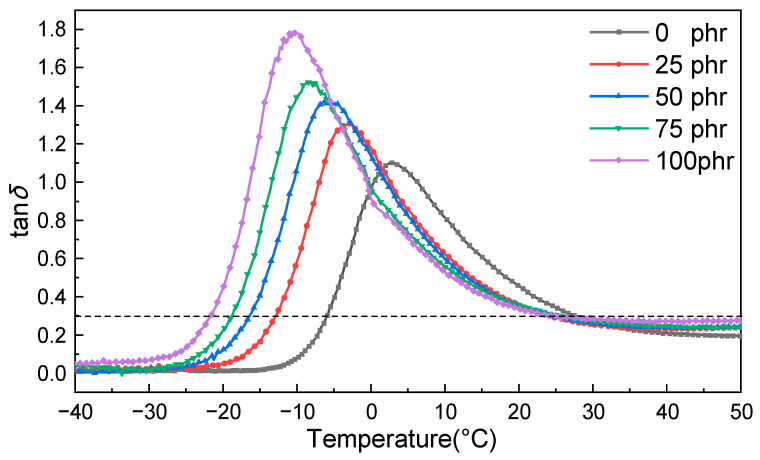
Effect of CP52 content on the damping performance of NBR.

**Figure 3 polymers-15-00604-f003:**
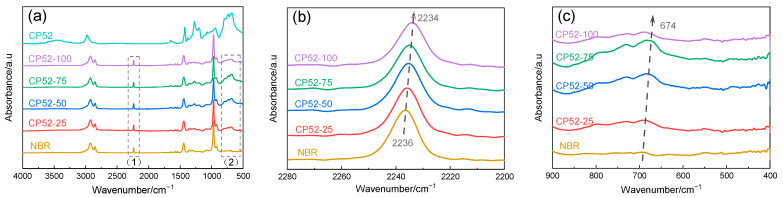
(**a**) FTIR spectra of CP52/NBR material (**b**) -CN (**c**) -Cl.

**Figure 4 polymers-15-00604-f004:**
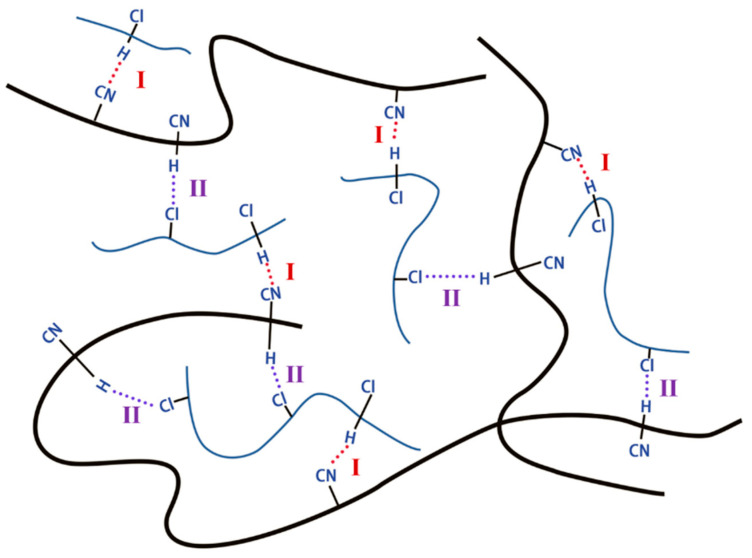
Schematic diagram of the CP52/NBR material network.

**Figure 5 polymers-15-00604-f005:**
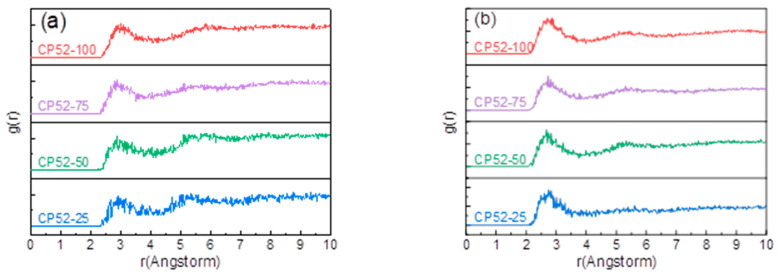
The radial distribution function (**a**) Cl-CP-H····NC-NBR. (**b**) CP-Cl····H-NBR-CN.

**Figure 6 polymers-15-00604-f006:**
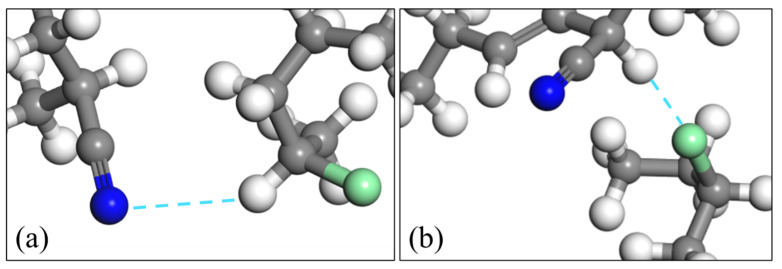
Schematic diagram of analogous hydrogen bonds in the CP52/NBR system. (**a**) Type I. (**b**) Type II.

**Figure 7 polymers-15-00604-f007:**
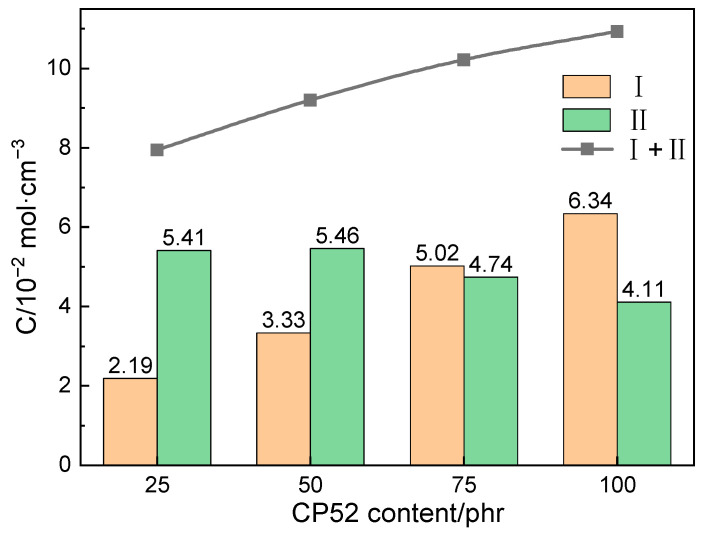
Molar concentrations of type I and type II analogous hydrogen bonds.

**Figure 8 polymers-15-00604-f008:**
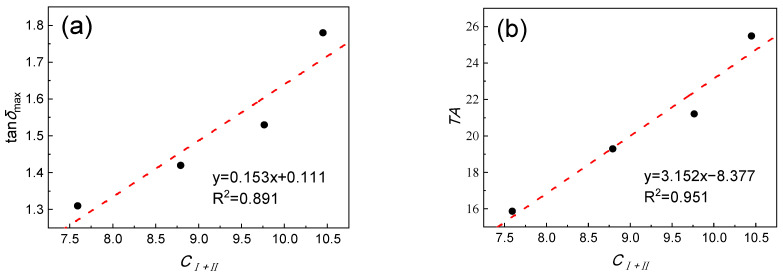
Linear regression curve. (**a**) *C_I+II_*-tan*δ*_max_. (**b**) *C_I+II_*-*TA.*

**Figure 9 polymers-15-00604-f009:**
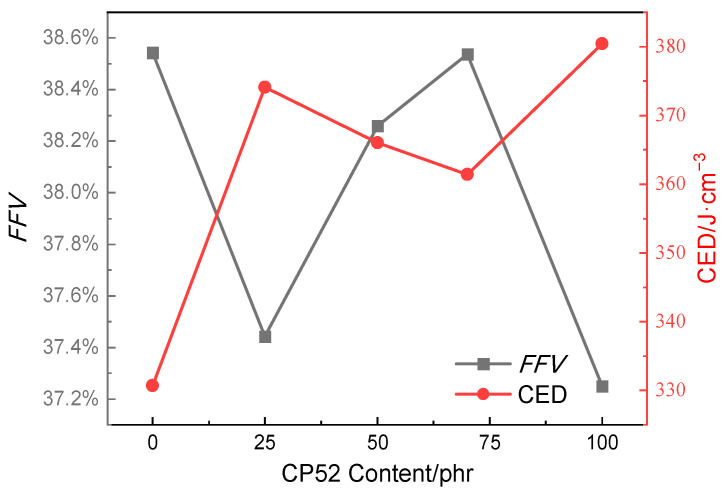
*FFV* and CED of CP52/NBR system.

**Table 1 polymers-15-00604-t001:** The number of CP52 and NBR molecules in the cell.

CP52/NBR	Mass Ratios of CP52	No. of NBR Molecules	No. of CP52 Molecules
0/100	0	4	0
25/100	20%	4	5
50/100	33.3%	4	11
75/100	42.8%	4	17
100/100	50.0%	4	23

**Table 2 polymers-15-00604-t002:** Densities by simulations and experiments.

CP52/NBR	*ρ*_MD_/g·cm^−3^	*ρ*_Exp_/g·cm^−3^	Relative Error
0/100	0.965	0.95	1.51%
25/100	1.018	1.02	−0.16%
50/100	1.042	1.03	1.19%
75/100	1.058	1.08	−2.07%
100/100	1.109	1.14	−2.73%

**Table 3 polymers-15-00604-t003:** Effect of CP52 content on the damping performance of NBR.

	CP52 Content/phr				
	0	25	50	75	100
*T*_g_/°C	1.81	−3.10	−6.40	−8.13	−10.50
tan*δ*_max_	1.10	1.31	1.42	1.53	1.78
Δ*T*/°C	33.7	37.0	40.6	44.3	46.8
*TA*	12.68	15.87	19.29	21.21	25.49

**Table 4 polymers-15-00604-t004:** *E_binding_* of the CP52/NBR system.

CP52/NBR	*E_total_* kcal/mol	*E_CP52_* kcal/mol	*E_NBR_* kcal/mol	*E_binding_* kcal/mol
0/100	277.24	0.00	277.24	0.00
25/100	24.722	−82.28	433.76	326.76
50/100	−205.59	−228.75	629.25	606.10
75/100	−515.21	−396.39	643.27	762.09
100/100	−858.61	−615.66	762.34	1005.29

## Data Availability

The data presented in this study are available upon request from the corresponding author.

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
