# Peer review of "Molecular-Level Insight of CP52/NBR Damping Composites through a Combination of Molecular Dynamics Simulation and Experimental Method"

_polymers, 2023, doi:10.3390/polym15030604_

Round 1
Reviewer 1 Report
A file is attached.

Reviewer 2 Report
1. Having in mind that the aim of the paper is to enhance the damping property of nitrile butadiene rubber (NBR), I believe it would be better to reorder your paper. For example:
· Page 1 (Abstract) line 9 – “To enhance the damping property of nitrile butadiene rubber (NBR), Chlorinated paraffin 52 (CP52) was blended with NBR to prepare CP52/NBR composites.” It would be better if you rewrite the sentence. For example: “To enhance the damping property of nitrile butadiene rubber (NBR), the elastomer used was blended with Chlorinated paraffin 52 (CP52) to prepare NBR/CP52 composites.
· Page 1 (Introduction) line 42 – “the small polar molecule CP52 was modified with NBR” is not correct. In fact the NBR was modified with CP52 in order to enhance its damping property. In my opinion, this should be corrected in the entire article.
2. In my opinion the Introduction section is to short and does not provide sufficient background to the paper. It is noted that there are few reports on using CP to enhance the damping property of rubber materials but these references are not cited in the paper. What makes your study different? What is the real aim of your study? Why did you choose to use NBR comprising 42% ACN instead of one comprising 18, 26 or 33% for example? Probably if you use NBR with lower ACN content (with lower glass transition temperature, respectively) the amount of the CP52 needed to improve the damping property of NBR will be smaller.
3. Page 2 (Materials and methods) line 59 – please, clarify whether the zinc oxide or magnesium oxide was used.
4. Page 2 (Materials and methods) lines 59 and 60 – In the scientific literature, the (2,2'-Dithiodibenzothiazole) as well as the tetramethyl thiuram disulfide are known as ACCELERATORS (not promoters) for sulfur vulcanization.
5. Page 2 (Materials and methods) line 67 – check the DMA description. I believe the heating rate was 3оC/min instead 3 K/min.
6. The results obtained are interesting. However, I think some additional results would be useful for the paper. What is the cross-link density (the extent of cross-linking)? The mechanical properties (tensile strength, elongation at break, Shore A hardness, etc) of NBR are not good enough in the absence of reinforcing fillers such as carbon black or silica. Considering that CP52 is commonly used as a secondary plasticizer, it is fully expected that it will also have an influence on the aforementioned characteristics, especially on the Shore A hardness of the studied composites. Are the Shore A hardness and the mechanical properties of the tested composites sufficient enough for vibration damping application?
Round 2
Reviewer 1 Report
The authors have answered to my scientific points, though I still think the trajectory time is not long. The English has been considerably improved. However, in lines 61-65, the imperative description is maintained, which is alright for a cooking recipe but not for a report of experimental procedures: "Then add", "hit", "hot-press and vulcanize". Overall, the paper can be now accepted with some minor improvements.
Author Response
We feel great thanks for your professional review work on our article. I have checked and corrected them on this occasion. I have also changed the tense of the section in lines 61-65 to the past tense, to ensure consistency with the rest of the text.

Reviewer 2 Report
I accept your comments
Author Response
We feel great thanks for your professional review work on our article.